# Pegivirus Detection in Cerebrospinal Fluid from Patients with Central Nervous System Infections of Unknown Etiology in Brazil by Viral Metagenomics

**DOI:** 10.3390/microorganisms12010019

**Published:** 2023-12-22

**Authors:** Rita de Cássia Compagnoli Carmona, Audrey Cilli, Antonio Charlys da Costa, Fabricio Caldeira Reis, Élcio Leal, Fabiana Cristina Pereira dos Santos, Bráulio Caetano Machado, Cristina Santiago Lopes, Ana Maria Sardinha Afonso, Maria do Carmo Sampaio Tavares Timenetsky

**Affiliations:** 1Enteric Disease Laboratory, Virology Center, Adolfo Lutz Institute, Sao Paulo 01246-900, Brazil; audreycilli@gmail.com (A.C.); fabriciocaldeirareis@hotmail.com (F.C.R.); braulio.machado@ial.sp.gov.br (B.C.M.); 2Institute of Tropical Medicine, University of São Paulo, Sao Paulo 05403-000, Brazil; charlysbr@yahoo.com.br; 3Institute of Biological Sciences, Federal University of Pará, Belem 66075-000, Brazil; elcioleal@gmail.com; 4Respiratory Disease Laboratory, Virology Center, Adolfo Lutz Institute, Sao Paulo 01246-900, Brazil; fabiana.santos@ial.sp.gov.br (F.C.P.d.S.); sanloppp@yahoo.com.br (C.S.L.); ana.afonso@ial.sp.gov.br (A.M.S.A.); 5Virology Center, Adolfo Lutz Institute, Sao Paulo 01246-900, Brazil

**Keywords:** metagenomic, pegivirus human, central nervous system infections, cerebrospinal fluid

## Abstract

Metagenomic next-generation sequencing (mNGS) methodology serves as an excellent supplement in cases where diagnosis is challenging to establish through conventional laboratory tests, and its usage is increasingly prevalent. Examining the causes of infectious diseases in the central nervous system (CNS) is vital for understanding their spread, managing outbreaks, and effective patient care. In a study conducted in the state of São Paulo, Brazil, cerebrospinal fluid (CSF) samples from 500 patients with CNS diseases of indeterminate etiology, collected between 2017 and 2021, were analyzed. Employing a mNGS approach, we obtained the complete coding sequence of *Pegivirus hominis* (HPgV) genotype 2 in a sample from a patient with encephalitis (named IAL-425/BRA/SP/2019); no other pathogen was detected. Subsequently, to determine the extent of this virus’s presence, both polymerase chain reaction (PCR) and/or real-time PCR assays were utilized on the entire collection. The presence of the virus was identified in 4.0% of the samples analyzed. This research constitutes the first report of HPgV detection in CSF samples in South America. Analysis of the IAL-425 genome (9107 nt) revealed a 90% nucleotide identity with HPgV strains from various countries. Evolutionary analyses suggest that HPgV is both endemic and extensively distributed. The direct involvement of HPgV in CNS infections in these patients remains uncertain.

## 1. Introduction

Human pegivirus, formerly named as persistent GB virus C (GBV-C) or hepatitis G virus, is a virus considered to be a lymphotropic virus and non-pathogenic [1,2,3,4]. It was first discovered in 1995 and initially named GBV-C due to its genetic similarities to the hepatitis C virus (HCV); it was later renamed as human pegivirus [3,5]. In contrast to HCV, human pegivirus is lymphotropic and sets up an asymptomatic infection [2,3,4].

The human pegivirus belongs to the genus *Pegivirus*, the *Flaviviridae* family, and consists of 11 species that infect a variety of hosts. Human pegivirus is represented by two species: *Pegivurus hominis* (HPgV) and *Pegivirus columbiaense*, termed human hepegivirus (HHPgV) or HPgV-2 [4,6,7]. HPgV is recognized as a virus with potentially diverse effects that vary depending on the clinical context and coinfection with other pathogens, although the clinical significance of these associations remains uncertain [8]. For instance, the virus has been detected in patients with various hematological disorders; there is evidence to suggest that HPgV may influence the outcomes of organ transplants and the response to immunosuppressive therapy. Additionally, studies have indicated that coinfection with HPgV may slow the progression to AIDS and prolong survival in HIV patients, likely due to immunological interactions between the two viruses. It may also provide a protective effect against the severity of liver diseases in co-infections with other hepatitis viruses [2,4,8].

HPgV has the potential to induce chronic infection, yet it remains uncorrelated with hepatitis or manifest clinical pathologies in ostensibly healthy individuals [2]. Numerous studies have reported a beneficial impact of HPgV infection on the progression of HIV disease. On the other hand, few investigations have established an association between HPgV infection and the manifestation of the disease [2,9]. Studies have been conducted since the characterization of HPgV illustrating that there is no causal relationship between the infection and any specific disease, suggesting a symbiotic or commensal interaction with the host [10]. The involvement of HPgV in patients with central nervous system (CNS) disorders has recently been questioned [11]. Although traditionally deemed non-pathogenic, some studies indicate it may have an impact on human health [1,12,13,14,15,16]. On the other hand, some studies have strongly supported the hypothesis of HPgV infection of the CNS [11].

HPgV possesses a 9.4 kb positive-sense single-stranded RNA (+ ssRNA) genomic structure. The genetic makeup of HPgV includes 5′ and 3′ untranslated areas (NTRs) and features an extensive open reading frame (ORF) encoding roughly 3000 amino acids, which undergo co- and post-translational divisions into four structural (S, X, E1, and E2) and six non-structural (NS) proteins [7]. The E2 glycoprotein of HPgV, instrumental in mediating adhesion and fusion with host cells, elicits the production of anti-HPgV antibodies. The main transmission routes of HPgV encompass blood exposure, sexual contact, intravenous substance use, and vertical transmission from mother to child, as well as exposure to infected blood and blood components [4,10]. Phylogenetic studies based on the analysis of the entire or partial genome (5’ UTR and/or E2 regions) of HPgV have been classified into seven genotypes with various subtypes. These genotypic classifications demonstrate distinct geographical distribution trends [6,9].

Initially identified as a blood-borne agent, HPgV has been detected in others biological matrices, including cerebrospinal fluid (CSF), prompting investigation into its potential involvement in CNS disorders. Studies addressing HPgV infection in patients with infectious diseases of the CNS have rarely been reported [1,12,13,14,15,16,17].

The nature of pegivirus pathogenicity and its possible impact on CNS diseases are still unknown. In Brazil, the limited studies conducted on the prevalence of HPgV have primarily been documented in healthy blood donors and patients infected with HIV-1 [18,19,20,21,22,23]. In this study, we report the detection of HPgV in patients with CNS infections of unknown etiology, and the complete coding sequence of the genome of one strain obtained from a patient with encephalitis using a metagenomic next-generation sequencing (mNGS) approach.

## 2. Materials and Methods

### 2.1. Sample Selection

This is a retrospective study conducted with convenience CSF samples. These samples were obtained from a laboratory surveillance program conducted in São Paulo State, Brazil, forwarded to the Virology Center at the Adolfo Lutz Institute Reference Laboratory for viral diagnosis under the Department of Health of the Government of São Paulo State, and were stored at a temperature of −80 °C. We analyzed 500 CSF samples collected between 2017 and 2021. They were derived from individuals suspected of having viral CNS diseases for which conventional diagnostic methods had not identified the causative agent. Specifically, real-time PCR assays for enterovirus, herpes virus group, varicella-zoster, respiratory viruses, and arbovirus proved negative. Sample inclusion criteria were applied to ensure a representative collection from cases where the causative agent remained unidentified by standard molecular techniques. Additionally, sample selection considered the availability and adequacy of specimens for analysis. Given the limitations of conventional diagnostics, we opted for a mNGS approach, a promising emerging tool, with the aim of clarifying the infectious CNS disease diagnoses with undefined etiologies. This strategy seeks to recognize a wider range of pathogens, including those not typically detected in standard protocols, thereby providing deeper insight into potential causes of the observed clinical manifestations.

### 2.2. Ethical Aspects

The study was carried out in accordance with the guidelines of the Declaration of Helsinki and received approval from the Ethics Committee of the Adolfo Lutz Institute, under Approval Number 5359914, dated 20 April 2022.

### 2.3. Metagenomic Sequencing

CSF samples were pooled based on the year of collection in groups of 5 samples. Viral particle-associated nucleic acids were enriched by filtration through a 0.45 μm filter (Millipore, Burlington, MA, USA) and filtrate was digested for 1 h at 37 °C with a mixture of nuclease enzymes consisting of TURBO DNase and RNase Cocktail Enzyme Mix (Thermo Fischer Scientific, Waltham, MA, USA), Baseline-ZERO DNase (Epicentre, Madison, WI, USA), Benzonase, (Merck Millipore, Billerica, MA, USA), and RQ1 RNase-Free DNase (Promega, Madison, WI, USA) to digest unprotected nucleic acids to reduce the content of non-encapsidated human nucleic acids. Nucleic acids then underwent automated extraction using the MagMAX Viral/Pathogen Nucleic Acid Isolation Kit (MVP II) (Thermo Fischer Scientific, Waltham, MA, USA). The cDNA synthesis was performed with a Superscript IV Reverse Transcriptase (Thermo Fisher Scientific, Waltham, MA, USA). A second-strand cDNA synthesis was performed using a DNA Polymerase I Large (Klenow) Fragment (Promega, Madison, WI, USA) and then purified with ProNex Size-Selective Purification System (Promega, Madison, WI, USA) and submitted to fluorometric quantification with QuantiFluor ONE dsDNA System (Promega, Madison, WI, USA). Subsequently, 10 ng of DNA was submitted to the Nextera XT Sample Preparation Kit (Illumina, San Diego, CA, USA) to construct a DNA library, which was identified using dual barcodes. For the size range, Pippin Prep (Sage Science, Inc., Beverly, MA, USA) was used to select a 500 bp insert (range 400–600 bp). The library was deep sequenced using a NovaSeq 6000 Sequencer (Illumina, San Diego, CA, USA) with 2 × 250 bp ends.

### 2.4. Bioinformatics

An analysis pipeline was used to analyze sequence data [24]. Briefly, we used taxonomy information to identify the human genome and bacterial and fungal sequences from the NCBI nt database for subtraction. Subsequently, two databases were generated: (1) a viral-specific BLASTx database assembled by incorporating viral protein sequences from the NCBI nr FASTA file, annotated under the virus kingdom in the taxonomic hierarchy, into the NCBI virus reference proteome; and (2) a non-redundant (NR) universal proteome database indexed utilizing the DIAMOND algorithm. In this database, each sequence was categorically labeled as either ‘virus’ or ‘non-virus’ based on taxonomic annotation, explicitly excluding sequences ascribed to the virus kingdom. Repeats and low-complexity regions were masked using segmasker from the blast+ suite (version 2.2.7) [25]. We developed an ensemble strategy that integrated the sequential use of various de Bruijn graph (DBG) and overlap-layout-consensus assemblers (OLC) with a novel partitioned sub-assembly approach called Ensemble Assemble [26]; contigs and reads were both used for sequence similarity search. The assembled contigs and singlets were aligned to our viral proteome database using BLASTx (version 2.2.7) with an E-value cutoff of <0.01. Matches to viral proteins were then aligned to our non-redundant (NR) universal proteome database using DIAMOND version 0.9.6 [25] to filter out non-viral hits that had better alignments to non-viral species. Reference sequences were annotated as virus or non-virus according to NCBI taxonomy in the DIAMOND NR database. When a read or contig was searched against this database using DIAMOND, this read was assigned to viruses or non-viruses according to the best hit to the reference sequence with the lowest E-value. If its best hit is non-virus, this read is removed from the final results.

### 2.5. Phylogenetic Analysis

Phylogenetic trees were constructed using the maximum likelihood approach, and branching support was estimated using an ultrafast bootstrap test with 1200 replications using the IQ-Tree tool [27]. Trees were visualized and edited using Figtree version 1.4.2 (http://tree.bio.ed.ac.uk/software/figtree, accessed on 14 September 2023).

### 2.6. Clinical and Epidemiological Data Source

Patient data encompassing demographics, clinical entities, and outcomes were extracted from the Laboratory Environment Management System (Gerenciador de Ambiente Laboratorial—GAL), overseen by the Ministry of Health (Rio Grande do Norte, Brazil). GAL, a computerized system tailored for public health laboratories, is utilized for conducting tests and assays on samples derived from human, animal, and environmental sources. Adhering to national standards, it is developed in compliance with the protocols established by the Brazilian Ministry of Health.

## 3. Results

### 3.1. HPgV Identification by mNGS

Here, we employed the mNGS methodology for the etiological determination of undiagnosed CNS disorders and successfully obtained the complete coding sequence of HPgV (9107 base pairs) from a CSF specimen (designated as IAL-425/BRA/SP/2019) of a patient presenting with suspected encephalitis. A total of 8206 reads were successfully mapped to the sequence, providing coverage of 206×. Notably, no additional pathogens were identified in this specimen.

### 3.2. Genotyping Tree

The IAL-425/BRA/SP/2019 strain was aligned with 100 full-length genome sequences available in the GenBank database. The nucleotide sequence determined in this study has been deposited in GenBank under the accession number OR639931. This genome is situated within a single, well-defined clade alongside HPgV references of species C (*Pegivirus hominis*), as visually depicted in Figure 1. This clade exhibits robust statistical support with a confidence level of 91% when compared to representative HPgV strains detected in China, France, Japan, South Africa, Spain, the United Kingdom, and the USA (Appendix A). Importantly, it is noteworthy that our analysis reveals a mixed distribution of HPgV sequences originating from various geographical regions, implying an absence of discernible geographic clustering within the phylogenetic tree. Additionally, our examination of the sequences within this specific clade, which is demarcated by a distinct blue area in the tree and includes the IAL-425/BRA/SP strain, shows a mean nucleotide identity of 90% within the polyprotein region. In contrast, the sequences outside of this clade, when compared to those within, exhibit a substantially lower mean nucleotide identity, falling below the 85% threshold.

In parallel, we present the phylogenetic tree constructed using genomic sequences of HPgV representing various genotypes obtained from GenBank (Appendix A). Noteworthy genotypes are explicitly identified within the tree. Our study’s sequence has been clustered within genotype 2 (Figure 2).

### 3.3. HPgV Detection

Subsequently, to confirm the presence of the virus and determine its prevalence across the entire collection, each of the 500 CSF samples was individually analyzed using a specific reverse transcription polymerase chain reaction (RT-PCR) assay and confirmed by partial sequencing of the conserved region of HPgV NS3 [28] (see Appendix A). Among the analyzed samples, HPgV was detected in 15 samples.

Sequencing of these 15 HPgV-positive samples generated sequences that ranged from 130 to 268 nucleotides in length. The average identity between these sequences was 86%. Thirty-one samples that tested negative in the RT-PCR assay and were part of a CSF pool with metagenomic reads were subjected to RT-qPCR for the detection of HPgV (see Appendix A). Among these analyzed samples, HPgV was detected in five samples.

The presence of HPgV was detected in 4.0% (*n* = 20/500) of the analyzed samples, inclusive of the IAL-425/BRA/SP/2019 strain. The 95% confidence interval for this prevalence ranged from approximately 2.28% to 5.72%.

Concerning CNS infections among HPgV-positive cases, 60.0% were attributed to aseptic meningitis, 25.0% to encephalitis, 10.0% to neurological syndrome, and 5.0% to acute flaccid paralysis, as outlined in Table 1. On the other hand, among patients with negative results for HPgV, the most commonly identified CNS infections in test requests were: aseptic meningitis, accounting for 59.2% of the cases; encephalitis (8.1%); meningoencephalitis (5.2%); acute flaccid paralyses (2.3%); and Guillain–Barré syndrome (1.3%). The remaining cases were more generically categorized as neurological syndromes, enteroviruses, or remained without detailed specification. The average age among the HPgV-negative and HPgV-positive groups was 31 and 26 years, respectively, with both groups having a median age of 26 years.

Only two positive cases for HPgV were documented with descriptive medical reports, designated as IAL-430/2017 and IAL-461/2019, respectively, as both provided medical reports detailing the progression of clinical conditions. The case description of Patient IAL-430/2017 involves a 22-year-old male with a history of alcohol consumption and injectable drug user (IDU). He presented with a symmetrical, progressive, and ascending decrease in muscle strength in the lower limbs. A CSF sample was taken, which showed slight protein–cell dissociation. The patient reported severe neuropathic leg pain. Magnetic resonance imaging revealed an extensive lesion in the upper dorsal spinal cord with spinal cord expansion, accompanied by enhancement of the medullary cone and nerve roots of the cauda equina. The primary diagnostic considerations were viral myelitis and neuroschistosomiasis. Treatment prescribed included glucocorticoids, antiviral medication, and broad-spectrum antiparasitic anthelmintics.

The case description of Patient IAL-461/2019 involves a 51-year-old male who developed encephalitis 13 days after undergoing a liver transplant due to alcoholic cirrhosis. The patient experienced daily fevers for 5 days prior to exhibiting severe encephalitic symptoms. Broad-spectrum antibacterial coverage was prescribed. In addition to the fever spikes, he displayed a deteriorating neurological status with myoclonus. CSF analysis was performed, which did not indicate bacterial meningitis. The CSF collection was conducted 2 days after the onset of symptoms to investigate viral agents.

This retrospective study aims to present the first report of HPgV in patients with CNS diseases in South America, and to broaden the understanding of the epidemiological and clinical characteristics associated with infection by this virus.

## 4. Discussion

In this study, the mNGS method was utilized to determine the etiological cause of undiagnosed CNS disorders from a laboratory surveillance program conducted in São Paulo State, Brazil, between 2017 and 2021. The application of mNGS in clinical diagnostics, particularly for undiagnosed CNS infections, has gained momentum in recent years due to its high sensitivity and broad range of pathogen detection [14,29,30].

The successful retrieval of a complete coding sequence of HPgV, spanning 9107 base pairs, from the IAL-425/BRA/SP/2019 strain from a patient with encephalitis is remarkable. HPgV is known to be lymphotropic [2,3,9], and its detection in CSF specimen and association with CNS infectious diseases presents an interesting avenue for further research, as the neurotropic potential of this virus is not widely documented. Furthermore, the absence of other pathogens in this specimen is noteworthy.

Numerous studies have reported a beneficial impact of HPgV infection on the progression of HIV disease and have been considered as therapeutic tools or viral vectors [2]. The pathogenic mechanisms and potential impact of HPgV on human health have not yet been confirmed [2].

In parallel, the detection of HPgV in 4.0% of the CSF samples in our study aligns with recent findings that suggest an increasing prevalence of this virus in diverse patient populations. While the pathogenicity of HPgV in CNS infections remains an area of ongoing investigation, its detection in varied clinical presentations underscores the need to consider it as a potential etiological agent in undiagnosed CNS cases. Among the general population and healthy blood donors, the estimative global prevalence of HPgV ranged from 0.8% to 44.6%, while in the at-risk population, the prevalence rate ranged from 1.8% to 75.3%. This prevalence has been higher in developing continents (e.g., Asia, Africa, and South America) than in the developed continents (e.g., North America, Europe, and Australia) [2].

The deep sequencing approach has been utilized in studies that detect HPgV in cases of CNS infections of unknown origin. In Poland, HPgV sequence was detected in the CSF and serum of three patients with encephalitis of unclear origin. The authors reported that these sequences from CSF differed from those circulating in serum. These findings are compatible with the presence of a separate viral compartment in the CNS [1]. In Australia, Kriesel et al. (2012) [12] detected HPgV in post-mortem brain tissue of a patient with multiple sclerosis, suggesting viral replication within the brain tissue. Fridholm et al. (2016) [14] reported the presence of a high viral load of human HPgV in the CSF of a Danish female suffering from severe encephalitis. No additional pathogens were found. The authors note that the pathogenicity of HPgV is uncertain, but it is noteworthy to detect a virus with a high viral load in the CNS.

The role of HPgV in patients with encephalitis and other CNS infectious diseases continues to be a subject of inquiry. On the other hand, recent studies have provided evidence supporting the potential of HPgV to infect the CNS [11,16]. Tuddenham et al. (2020) [16] described the detection of HPgV as the sole pathogen in ante-mortem cerebral tissue, cerebrospinal fluid, and serum of a patient with encephalitis. Valyraki et al. (2023) [11] reported one case of an immunocompromised patient with severe myelitis and three cases with optic neuropathy causing blindness. Extensive investigation yielded negative results, but CSF analysis using the metagenomics approach was positive for HPgV. These studies reinforce that the clinical significance of HPgV detection in samples of brain tissue, CSF, and serum warrant further investigation to confirm their potential as neurotropic viruses.

Some viruses belonging to the *Flaviviridae* family, such as the West Nile virus, tick-borne encephalitis virus, dengue virus, and Zika virus, exhibit neurotropism and contribute substantially to encephalitis cases in Europe, North America, and South America. As HPgV also belongs to this family, its neurotropic potential has been under consideration [1,31,32]. Interestingly, our study’s breakdown of clinical syndromes among HPgV-positive cases offers insight into its potential clinical manifestations. In this study, aseptic meningitis constituted the majority of HPgV-positive cases, representing 60.0% of the total. The occurrence of aseptic meningitis among patients was almost the same regardless of whether they were infected with HPgV or not. This suggests that HPgV may not be a major differentiating factor in the incidence of aseptic meningitis among these patients. Aseptic meningitis is commonly associated with various viral agents, especially enteroviruses [33]. On the other hand, our findings suggest that HPgV could also play a role in the development of this disease.

The clinical information for this study was exclusively obtained from the requisitions, accompanied by the CSF samples and the records in the GAL system. These requisitions and records contain limited information, focusing primarily on basic demographic data (age, sex, and municipalities of residence) and clinical reasons for testing (clinical suspicion, onset of symptoms date, and collection date). While we recognize the importance of investigating other potential CNS diseases, our data collection was confined to the information already recorded in the available samples. We acknowledge the limitation of not including more extensive clinical data. Despite these limitations, we reiterate the importance of the detection of pegivirus in the CSF of patients with CNS diseases, albeit based on a limited data set, which offers a new perspective that merits further investigation. Our study is the first to explore this association in South America and could be a preliminary step toward future research in Brazil, where access to more detailed and comprehensive data might be possible. We were able to obtain the complete medical records of only two patients who tested positive for HPgV. Two cases described in this study, IAL-430/2017 and IAL/461/2019, shed light on the varied and multifaceted clinical presentations associated with HPgV infection. Both cases offer intriguing insights into the possible neurological complications of the virus. For patient IAL-430/2017, their case report details the incidence of viral myelitis, a condition wherein the spinal cord becomes inflamed due to viral infections [34]. Historically, various viruses, including viruses from the *Picornaviridae* family, herpesviruses, and flaviviruses, have been identified as etiological agents for viral myelitis [34,35,36]. Patient IAL-461/2019 presents a more complex clinical scenario, with the onset of encephalitis occurring post-liver transplant. Transplant patients, due to immunosuppression, are more susceptible to a range of infections, including those of viral origin [8,37,38]. Due to shared transmission routes (parenteral and/or sexual), HPgV viremia is even higher in individuals infected with HIV, HCV, or who are at risk for such infections, such as IDU or patients undergoing procedures such as hemodialysis, blood transfusion, and organ transplantation [39,40].

In this study, the phylogenetic analysis demonstrated that the HPgV strain IAL-425/BRA/SP/2019 is within a well-defined singular clade and highlights its genetic relationship with HPgV references of the species *Pegivirus hominis*. This analysis found no such geographic clustering for HPgV. This suggests a broad dispersal pattern for HPgV across different regions worldwide, which may be influenced by global human movement or other transmission dynamics [41].

The analysis of sequences placed the Brazilian isolate firmly into the genotype 2 clade. The distribution of HPgV genotypes has demonstrated varying patterns of distribution worldwide. Genotypes 1 and 2 of HPgV are primarily found in Africa, with genotype 2 being more prevalent in Europe. Genotype 3 predominates in Asian countries and South America, while genotypes 4 and 5 continue to be dominant in the Philippines and other Southeast Asian countries. Genotype 6 is circulating in Indonesia. Genotype 7 was recently discovered in the Yunnan Province of China, as well as in other Asian countries, including Qatar. The differences in the distribution of HPgV types may be associated with the origin, evolution, and transmission of these genotypes [2,9,42,43].

The predominance of HPgV genotype 2 is well documented in Brazil, followed by genotypes 1, 3, and 5, having been described in different regions of the country, including the North, Northeast, Central-West, Southeast, and South, and in diverse populations, such as transplant recipients’ kidney disease [38], blood donors [19,44,45], HIV-infected individuals [45,46], individuals with chronic hepatitis C [47,48], and polytransfused individuals with thalassemia [22]. On the other hand, this finding represents the first description of HPgV in CSF samples from patients with CNS infection in Brazil and South America. Due to its vast territory and climatic diversity, Brazil provides a potential setting for the variation of HPgV genotypes. However, there is still a lack of studies within the country to explore and confirm this relationship, underscoring a need for research in this field.

## 5. Conclusions

This study represents the first documentation of HPgV in CSF samples from patients with CNS disorders in South America. HPgV is frequently classified as a commensal and persistent virus in various studies, associated with asymptomatic human infections [10,14,40]. Despite the low prevalence observed in this study, the identification of the complete coding sequence of the viral genome in a sample warrants special attention. The discovery is noteworthy, suggesting a potential etiological agent for some previously unidentified CNS diseases. Moreover, the effectiveness of the metagenomics approach in identifying unknown pathogens is underscored. However, the study is limited by the absence of clinical and epidemiological data for patients who tested positive for HPgV, which constrains our ability to link the virus to CNS disorders. These findings emphasize the urgent need for more comprehensive investigations to elucidate the role of this virus in CNS infectious diseases and to explore its potential clinical and therapeutic implications, particularly in the Brazilian setting.

## Figures and Tables

**Figure 1 microorganisms-12-00019-f001:**
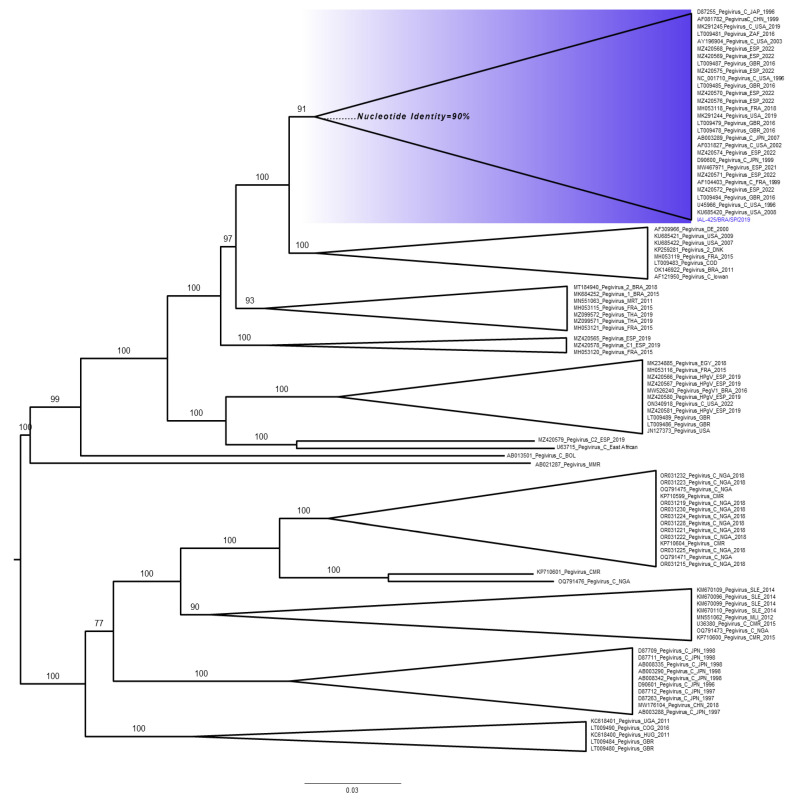
Phylogenetic analysis of HPgV: A phylogenetic tree was constructed using complete genome sequences of pegiviruses retrieved from GenBank, with GenBank IDs provided for each sequence. The sequence from our study, labeled as IAL-425/BRA/SP/2019, is included. For easier visualization, clusters have been condensed, with numbers in each indicating the statistical support of these clades, obtained through 1200 bootstrap replications. The clade containing the complete coding sequence of genome generated in this study is highlighted in blue. Horizontal bars at the base of the tree represent a scale indicating nucleotide substitutions per site.

**Figure 2 microorganisms-12-00019-f002:**
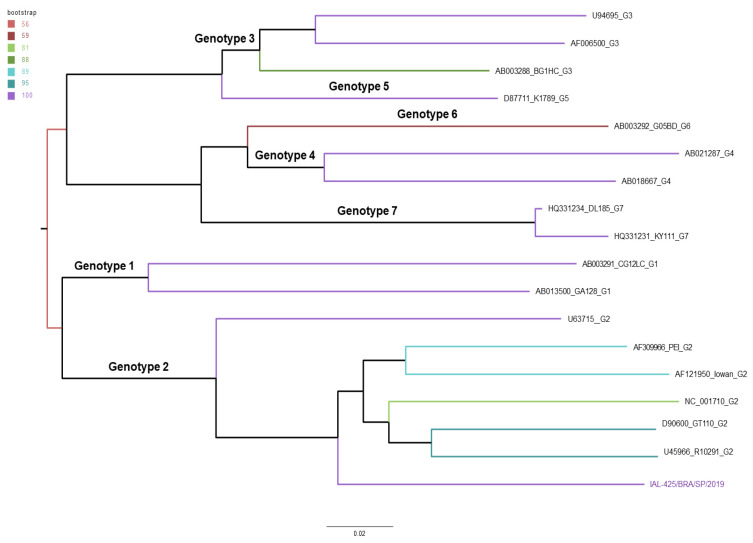
Phylogenetic analysis of HPgV genotypes. Phylogenetic tree was constructed using genomic sequences of HPgV representing distinct genotype reference strains obtained from GenBank. Our study’s sequence is denoted as IAL-425/BRA/SP/2019. To ensure the robustness of our analysis, we employed 1200 bootstrap replications, providing statistical support for the tree’s structure. Branch colors on the tree correspond to a chart indicating the level of support obtained from the bootstrapping process. At the base of the tree, horizontal bars serve as a scale representing nucleotide substitutions per site.

**Table 1 microorganisms-12-00019-t001:** Summary of clinical and epidemiological details of detected cases of HPgV from 2017 to 2021, in São Paulo, Brazil.

Case	Lab ID	Year	State	Municipality	Onset Date	Collection Date	Sex	Age		Clinical Suspicion
1	IAL-224	2017	SP	Mogi das Cruzes	19/05/2017	23/05/2017	M	3	years	Encephalitis
2 ^a^	IAL-430	2017	SP	São Paulo	05/12/2017	N.A.	M	22	years	Myelitis/Aseptic Meningitis
3	IAL-024_4	2018	SP	São Paulo	16/05/2018	16/05/2018	M	34	years	Encephalitis
4	IAL-026_3	2018	SP	São Paulo	20/05/2018	21/06/2018	M	26	years	Aseptic Meningitis
5	IAL-028_5	2018	SP	São Paulo	N.A.	11/09/2018	M	75	years	Neurological syndrome
6	IAL-029_1	2018	SP	Campinas	N.A.	19/09/2018	F	35	years	Aseptic Meningitis
7	IAL-108	2018	SP	São Paulo	10/03/2018	10/03/2018	M	2	years	Aseptic Meningitis
8	IAL-206	2018	SP	São Paulo	06/03/2018	06/04/2018	M	3	years	Aseptic Meningitis
9	IAL-666	2018	SP	Mogi das Cruzes	04/08/2018	04/08/2018	F	5	years	Aseptic Meningitis
10	IAL-670	2018	SP	São Paulo	25/07/2028	08/08/2018	M	1	years	Acute Flaccid Paralysis
11	IAL-073	2019	SP	Mauá	05/03/2019	05/03/2019	F	26	years	Aseptic Meningitis
12	IAL-187	2019	SP	Mogi das Cruzes	31/03/2019	06/04/2019	F	1	years	Aseptic Meningitis
13 ^b^	IAL-425	2019	SP	Campinas	16/06/2019	03/07/2019	M	65	years	Encephalitis
14 ^c^	IAL-461	2019	SP	Botucatu	15/07/2019	17/07/2019	M	51	years	Encephalitis
15	IAL-485	2019	SP	Mogi das Cruzes	03/08/2019	04/08/2019	M	27	years	Aseptic Meningitis
16	IAL-548	2019	SP	São Paulo	21/09/2019	22/09/2019	M	10	months	Aseptic Meningitis
17	IAL-221	2020	SP	São Paulo	15/08/2020	01/09/2020	M	39	years	Neurological syndrome
18	IAL-048	2021	SP	Campinas	16/01/2021	02/03/2021	M	52	years	Encephalitis
19	IAL-106	2021	SP	Mogi das Cruzes	08/05/2021	10/05/2021	M	39	years	Aseptic Meningitis
20	IAL-084_4	2021	SP	São Paulo	01/05/2021	18/05/2021	F	17	years	Aseptic Meningitis

^a^ illicit drug user; ^b^ complete coding sequence of HPgV; ^c^ liver transplant recipient; N.A.: not available. Date: day/month/year.

## Data Availability

The consensus sequence of the viruses analyzed in this study was submitted to the GenBank database under the accession number OR639931.

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
