# Peer review of "Pegivirus Detection in Cerebrospinal Fluid from Patients with Central Nervous System Infections of Unknown Etiology in Brazil by Viral Metagenomics"

_microorganisms, 2023, doi:10.3390/microorganisms12010019_

Round 1

Reviewer 1 Report

Comments and Suggestions for Authors

Human pegivirus-1 (HPgV-1) infection in the brain has not been extensively examined and its association with disease remains unconfirmed. The author reports the detection of HPgV-1 in patients with central nervous system (CNS) infections of unknown etiology and presents the complete genome sequence of one strain obtained from a patient with encephalitis using a metagenomic next-generation sequencing (mNGS) approach. This research is relatively new, and I have the following comments:

1. Please add more introduction on the HPgV-1 virus, especially its clinical significance, including in other site besides CNS.

2. The samples were from those suspected of CNS viral infection. Does these patients also exluded other possible CNS disease, e.g.,  autoimmune encephalitis.

3. Please collect the clinical charicteristics of the samples.  Is there any difference between the HPgV-1 postive and the negative groups.

4. The authors may also test the HPgV-1 in the blood of the samples from these subject, and could benefit from comparing the inconsistency between blood and CSF.

5. Add the 95% confidential intervals of the prevalence.

6. Could the authors have the outcome of the patients?

Author Response

We sincerely thank you for your constructive and detailed suggestions. Your guidance has been crucial in improving the manuscript. Below are the responses to your recommendations:

1. Please add more introduction on the HPgV-1 virus, especially its clinical significance, including in other site besides CNS.

Authors: We are grateful for pointing out this gap in our manuscript. Based on your recommendation, I have expanded the introductory section to include additional details about the HPgV. Please, refer to lines 46 to 55 in the revised manuscript R1.

2. The samples were from those suspected of CNS viral infection. Does these patients also exluded other possible CNS disease, e.g.,  autoimmune encephalitis.

Authors: This indeed represents a limitation of our study, as described in the discussion section. Due to the lack of comprehensive clinical data on the patients, we do not have additional clinical information to confirm or exclude the presence of other CNS diseases. This limitation has been acknowledged in our study to ensure a transparent and accurate understanding of the context and scope of our findings. Please, refer to lines 381 to 383 in the revised manuscript

3. Please collect the clinical charicteristics of the samples.  Is there any difference between the HPgV-1 postive and the negative groups.

Authors: Thank you for your suggestions. We have included the clinical characteristics of the samples in the results and discussion sections. Please, refer to lines 238 to 245; and 327 to 330 in the revised manuscript R1.

4. The authors may also test the HPgV-1 in the blood of the samples from these subject, and could benefit from comparing the inconsistency between blood and CSF.

Authors: We appreciate the suggestion to test HPgV-1 in the blood samples. However, we do not have access to blood samples from these patients, so this comparison is not possible with our current data.

5. Add the 95% confidential intervals of the prevalence.

Authors: Thank you for your suggestion. We have incorporated these calculations into the results. Please, refer to lines 234 to 235 in the revised manuscript R1.

6. Could the authors have the outcome of the patients?

Authors: Thank you for your question. In response, we only possess the negative results for the viral agents tested, as outlined in the materials and methods section. We do not have additional outcome data for the patients.

Reviewer 2 Report

Comments and Suggestions for Authors

The study of de Cássia Compagnoli Carmona et al. reports the detection of a nearly (?) full-length genome of human pegivirus (HPgV) genotype 2 in a CSF sample of a patient with encephalitis using viral metagenomics as well as the detection of additional 19 HPgV-1 positive CSF samples using RT-PCR/RT-qPCR techniques. The manuscript is straightforward and overall well written, although some modifications should be done to improve its scientific quality.

Major points:

1    . The binomial nomenclature of family Flaviviridae including the genus Pegivirus was officially introduced recently (https://ictv.global/report/chapter/flaviviridae/flaviviridae/pegivirus, Simmonds, P., Becher, B., Bukh, J., Gould, E.A., Meyers, G., Monath, T., Muerhoff, S., Pletnev, A., Rico-Hesse, R., Smith, D.B., Stapleton, J.T., and ICTV Report Consortium. 2017, ICTV Virus Taxonomy Profile: Flaviviridae, Journal of General Virology, 98:2–3.) which should be adopted to this study (including in the figures) as well. E.g.: species Pegivirus C was renamed as Pegivirus hominis, while species Pegivirus H now called as Pegivirus columbiaense. The virus name is human pegivirus genotype 1-7.

2.      Some descriptions of the applied RT-PCR and RT-qPCR reaction conditions should be also included to the Materials and Methods section of the main text with reference to the supplementary (the description of applied RT-qPCR assay found among the supplementary files was detailed enough but similar description should be created to the RT-PCR screening reactions as well).

3.      It should be considered to acquire additional sequence data from the additional HPgV positive samples or at least make a phylogenetic analysis (or at least give pairwise sequence identities) of the 15 partial HPgV NS3 sequences determined by RT-PCR screening reactions (line 216). The sequence divergence (if there is any) between the identified HPgV strains should be analyzed and discussed.

4.      I understand that there was no additional pathogen detected by NGS in the sample pool which contained the index HPgV-2, but was there any other pathogen detected by NGS in the sample pools which contains the additional 19 HPgV RT-PCR/qPCR positive samples? If yes, then it should be included to the manuscript (maybe should be summarized in a table format).

5.      The Figure 1 should be modified: The upper part of the figure seems blurry, hard to read. Because the authors discuss the geographical origins of HPgVs (lines 180-181) the names of the countries of different HPgVs should be included to the figure. The genotypes of the different strains also should be included here. The figure legends of Figure 1 contain that “A phylogenetic tree was constructed using complete genomes of pegiviruses viruses retrieved from Genbank.” (lines 190-191), but as I see the tree contains only human strains so either the tree should be supplemented with pegiviruses from animals, or the legends should be re-written accordingly.

6.      The authors mentioned the “considerable viral load” of the HPgV-2 (line 353). Some experimental data should be added here (absolute quantification of the viral load using the available RT-aPCR assay, or at least a Cq values from the RT-qPCR reaction) to support the assumption. 

Minor points:

Lines 23-24: The term “full-nearly complete” is non-sense. Full length genome (e.g. line 194) or nearly complete (line 165)? Please specify that which part of the genome is missing (as well as in line 165).

Lines 172-173: The sentence should be re-phrased (hard to understand)

Lines 189-195: The figure legend (as well as the manuscript) should be proofread by a native English speaker. The sentences “Phylogenetic of HPgV-1.” and “To facilitate visualization cluster were collapsed.” seems incorrect to me.

Lines 206-207: What does it mean “Noteworthy genotypes are explicitly identified within the tree”? It should be clarified.

Comments on the Quality of English Language

Some parts of the manuscript nedds to be checked by a native englis speaker

Author Response

We sincerely thank you for your constructive and detailed suggestions. Your guidance has been crucial in improving the manuscript. Below are the responses to your recommendations:

Major points:

1. The binomial nomenclature of family Flaviviridae including the genus Pegivirus was officially introduced recently (https://ictv.global/report/chapter/flaviviridae/flaviviridae/pegivirus, Simmonds, P., Becher, B., Bukh, J., Gould, E.A., Meyers, G., Monath, T., Muerhoff, S., Pletnev, A., Rico-Hesse, R., Smith, D.B., Stapleton, J.T., and ICTV Report Consortium. 2017, ICTV Virus Taxonomy Profile: Flaviviridae, Journal of General Virology, 98:2–3.) which should be adopted to this study (including in the figures) as well. E.g.: species Pegivirus C was renamed as Pegivirus hominis, while species Pegivirus H now called as Pegivirus columbiaense. The virus name is human pegivirus genotype 1-7.

Authors: Thank you for your suggestions. We have included the latest classification of pegivirus in the introduction and throughout the article. Please, refer to lines 43 to 46 in the revised manuscript R1.

2. Some descriptions of the applied RT-PCR and RT-qPCR reaction conditions should be also included to the Materials and Methods section of the main text with reference to the supplementary (the description of applied RT-qPCR assay found among the supplementary files was detailed enough but similar description should be created to the RT-PCR screening reactions as well).

Authors: Thank you for your suggestion. We have decided to provide a concise description of the RT-PCR methodology and also including in the Supplementary Material, similar to what we have done with the RT-qPCR description. 

3. It should be considered to acquire additional sequence data from the additional HPgV positive samples or at least make a phylogenetic analysis (or at least give pairwise sequence identities) of the 15 partial HPgV NS3 sequences determined by RT-PCR screening reactions (line 216). The sequence divergence (if there is any) between the identified HPgV strains should be analyzed and discussed.

Authors: We appreciate your valuable comments. We conducted the sequencing of 15 HPgV-positive samples, generating sequences ranging from 130 to 268 nucleotides in length. The average identity among these sequences is 86%. Subsequently, we aligned these partial sequences with the corresponding regions of HPgV references to construct a maximum likelihood tree. However, the resulting tree lacked resolution, as references of the same genotype did not cluster together. This lack of resolution was further confirmed through maximum likelihood mapping analysis, revealing that only 52% of the trees in this alignment were resolved. This stands in stark contrast to the 95% resolution observed in the alignment of complete genomes, which was utilized to genotype our full-length HPgV sequence. Due to these limitations, we chose not to include additional details about the partial sequences in the current manuscript.

4. I understand that there was no additional pathogen detected by NGS in the sample pool which contained the index HPgV-2, but was there any other pathogen detected by NGS in the sample pools which contains the additional 19 HPgV RT-PCR/qPCR positive samples? If yes, then it should be included to the manuscript (maybe should be summarized in a table format). 

Authors: We appreciate your attention to detail. Although other novel viruses (eg. Botybirnavirus, Gammapartitivirus and Totivirus) were detected in some samples alongside Pegiviruses, no further analysis was conducted, primarily due to their relatively small size. We are concentrating on these discoveries and are in the process of designing new primers for future confirmation by RT-PCR. Therefore, this article specifically focuses on reporting the findings and confirmation of HPgV detections.

5. The Figure 1 should be modified: The upper part of the figure seems blurry, hard to read. Because the authors discuss the geographical origins of HPgVs (lines 180-181) the names of the countries of different HPgVs should be included to the figure. The genotypes of the different strains also should be included here. The figure legends of Figure 1 contain that “A phylogenetic tree was constructed using complete genomes of pegiviruses viruses retrieved from Genbank.” (lines 190-191), but as I see the tree contains only human strains so either the tree should be supplemented with pegiviruses from animals, or the legends should be re-written accordingly.

Authors: Thank you for your suggestions, which we have carefully considered and addressed. As requested, we have added the countries of origin and rewritten the figure legend for enhanced clarity. When available in Blast, genotypes have been included in the phylogenetic tree. For example, Pegivirus C, Pegivirus C1.

6. The authors mentioned the “considerable viral load” of the HPgV-2 (line 353). Some experimental data should be added here (absolute quantification of the viral load using the available RT-aPCR assay, or at least a Cq values from the RT-qPCR reaction) to support the assumption.

Authors: Thank you for your question. We have decided to remove the phrase "presumably with a considerable viral load”. We recognize that this assumption was not backed by direct evidence, as the absolute quantification of viral load was not analyzed and fell outside the main scope of our study. This modification ensures that our conclusions are grounded only in the data and analyses we conducted. Please, refer to lines 377 to 378 in the revised manuscript R1.

Minor points:

Lines 23-24: The term “full-nearly complete” is non-sense. Full length genome (e.g. line 194) or nearly complete (line 165)? Please specify that which part of the genome is missing (as well as in line 165).

Authors: Thank you for your observation. Although certain segments at the extremities are absent, the coding region of the Pegivirus is fully. As a result, we are justified in referring to it as a 'full-length genome' in our manuscript.

Lines 172-173: The sentence should be re-phrased (hard to understand)

Authors: Thank you for pointing out the need for clarification. The sentence has been revised for better comprehension. Please, refer to lines 183 to 184 in the revised manuscript R1.

Lines 189-195: The figure legend (as well as the manuscript) should be proofread by a native English speaker. The sentences “Phylogenetic of HPgV-1.” and “To facilitate visualization cluster were collapsed.” seems incorrect to me.

Authors: Thank you for your suggestion. We have revised the text for clarity in English and replaced 'collapsed' with 'condensed' as a synonym to enhance reader comprehension. This adjustment should improve the clarity of our description. Please, refer to lines 201 to 207 in the revised manuscript R1.

Lines 206-207: What does it mean “Noteworthy genotypes are explicitly identified within the tree”? It should be clarified.

Authors: Thank you for your question. Our intention was described "Distinct genotypes of references strains are identified within the tree".  We have reformulated the figure legend to enhance clarity and comprehension. Please, refer to lines 217 in the revised manuscript R1.

Round 2

Reviewer 1 Report

Comments and Suggestions for Authors

Thank you to the author for their response and for revising certain manuscript sections. I still have the following suggestions:

1. Table 1 of this article only provides simple clinical information, including gender, age, and clinical suspicion. The author cannot provide relevant information on whether this group of patients has ruled out other possible central nervous systems(CNS) diseases, such as autoimmune encephalopathy. However, the author indicated that this part of the data is not available, but this will reduce the credibility of the role of HPgV virus infection in CNS diseases in this group of patients.

2. The author mentioned that 60.0% of CNS viral infections among HPgV-positive cases were attributed to aseptic meningitis. However, later in the article, it is stated that the incidence of aseptic meningitis was essentially the same regardless of HPgV infection status. Therefore, HPgV may not be a significant differentiator for aseptic meningitis in these patients. Additionally, the article lacks treatment and prognosis data, and there is no definitive evidence indicating a relationship between the virus and HIV combined with CNS infections.

3. The author's response to question 3 can be located in lines 238 to 245 of the article and 327 to 330 in the revised manuscript R1. However, this section currently encompasses only age, gender, and clinical suspicion of the disease. Collecting the clinical characteristics of the samples, including clinical symptoms, physical examination, and laboratory examination data, is recommended. Furthermore, it is suggested to provide additional analysis regarding the differences between the HPgV-1 positive and negative groups.

Author Response

Thank you to the author for their response and for revising certain manuscript sections. I still have the following suggestions:

1. Table 1 of this article only provides simple clinical information, including gender, age, and clinical suspicion. The author cannot provide relevant information on whether this group of patients has ruled out other possible central nervous systems(CNS) diseases, such as autoimmune encephalopathy. However, the author indicated that this part of the data is not available, but this will reduce the credibility of the role of HPgV virus infection in CNS diseases in this group of patients.

2. The author mentioned that 60.0% of CNS viral infections among HPgV-positive cases were attributed to aseptic meningitis. However, later in the article, it is stated that the incidence of aseptic meningitis was essentially the same regardless of HPgV infection status. Therefore, HPgV may not be a significant differentiator for aseptic meningitis in these patients. Additionally, the article lacks treatment and prognosis data, and there is no definitive evidence indicating a relationship between the virus and HIV combined with CNS infections.

Authors 1 and 2 questions: Thank you for your valuable observations of our article. We appreciate the importance of providing a comprehensive clinical context and treatment and prognosis data to enhance the credibility of our findings. However, we would like to clarify some points that may help in understanding the inherent limitations of our study design and the significance of our findings within these constraints.

Our retrospective study was conducted using convenience samples. We include the information in the material and methods section (highlighted in blue). The clinical information for this study was exclusively obtained from the requisitions accompanying the CSF samples and the records in the GAL system. These requisitions and records typically contain limited information, focusing primarily on basic demographic data and clinical reasons for testing. While we recognize the importance of investigating other potential CNS diseases, our data collection was confined to the information already recorded in the available samples.

In the article, we explicitly acknowledge the limitation of not including more extensive clinical data. This limitation was mentioned in the conclusion section to ensure the study's credibility, transparency, and to enable readers to evaluate the findings in the appropriate context.  We also added a paragraph to the discussion describing in more detail the limitation of our study. The modifications made in the current version are highlighted in blue

Despite these limitations, we reiterate the importance of our findings. The detection of pegivirus in the CSF of patients with CNS diseases, albeit based on a limited data set, offers a new perspective that merits further investigation. Our study “suggests,” rather than confirms, a possible link between the HPgV virus and CNS diseases. It is one of the first studies to explore this association in South America and could be a preliminary step toward future research in Brazil, where access to more detailed and comprehensive data might be possible.

3. The author's response to question 3 can be located in lines 238 to 245 of the article and 327 to 330 in the revised manuscriptR1. However, this section currently encompasses only age,gender, and clinical suspicion of the disease. Collecting the clinical characteristics of the samples, including clinicalsymptoms, physical examination, and laboratory examinationdata, is recommended. Furthermore, it is suggested to provideadditional analysis regarding the differences between the HPgV-1 positive and negative groups.

 Authors:  We appreciate your recommendation.

As mentioned in our manuscript, the clinical data we had access to be confined to age, sex, and clinical suspicion of the disease, obtained from the requisitions accompanying the CSF samples and the GAL system records. Unfortunately, detailed clinical symptoms, physical examination results, and comprehensive laboratory data were not available for our retrospective analysis.

Given these constraints, the only feasible comparisons we could conduct were based on clinical suspicion, age and gender between the HPgV-positive and negative groups, as as described throughout the text. While we acknowledge the value that a more extensive analysis of clinical characteristics would add to our study, such an analysis was beyond the scope of our available data.  This study aimed to present some evidence of HPgV virus involvement in CNS diseases to instigate further investigations to clarify its role in the pathogenesis of CNS disorders.

We have endeavored to be transparent about these limitations in our manuscript, especially in discussing the implications of our findings and their potential application in future research. 

We hope that the revised manuscript, with its focus on the available data and analysis, is found to be a valuable contribution to the field and meets the criteria for publication in Microorganisms.

Thank you again for your consideration and valuable feedback.

Reviewer 2 Report

Comments and Suggestions for Authors

If the sequence of IAL-425/BRA/SP/2019 is not complete (as stated by the authors in the answers "certain segments at the extremities are absent") please consider using the term "complete-coding sequence" or CDS throughout the text. If it is not complete, then you should not refer as "full-length".

Please consider including a section of your answers ("We conducted the sequencing of 15 HPgV-positive samples, generating sequences ranging from 130 to 268 nucleotides in length. The average identity among these sequences is 86%.") into the main text. I think it is a result worth to publish it.

Author Response

If the sequence of IAL-425/BRA/SP/2019 is not complete (as stated by the authors in the answers "certain segments at the extremities are absent") please consider using the term "complete-coding sequence" or CDS throughout the text. If it is not complete, then you should not refer as "full-length".

Authors: Thank you for your suggestions. We have made the modifications throughout the text as suggested. The modifications made in the current version are highlighted in blue.

Please consider including a section of your answers ("We conducted the sequencing of 15 HPgV-positive samples, generating sequences ranging from 130 to 268 nucleotides in length. The average identity among these sequences is 86%.") into the main text. I think it is a result worth to publish it.

Authors: We would like to extend our sincere thanks for your pertinent suggestion, which have will enrich our article. We have include the data in the results section. The modifications made in the current version are highlighted in blue.